# A Structure-Based View on ABC-Transporter Linked to Multidrug Resistance

**DOI:** 10.3390/molecules28020495

**Published:** 2023-01-04

**Authors:** Jiahui Huang, Gerhard F. Ecker

**Affiliations:** Department of Pharmaceutical Sciences, University of Vienna, 1140 Vienna, Austria

**Keywords:** structure-based studies, ABC transporter, multidrug resistance (MDR), cancer therapy

## Abstract

The discovery of the first ATP-binding cassette (ABC) transporter, whose overexpression in cancer cells is responsible for exporting anticancer drugs out of tumor cells, initiated enormous efforts to overcome tumor cell multidrug resistance (MDR) by inhibition of ABC-transporter. Because of its many physiological functions, diverse studies have been conducted on the mechanism, function and regulation of this important group of transmembrane transport proteins. In this review, we will focus on the structural aspects of this transporter superfamily. Since the resolution revolution of electron microscope, experimentally solved structures increased rapidly. A summary of the structures available and an overview of recent structure-based studies are provided. More specifically, the artificial intelligence (AI)-based predictions from AlphaFold-2 will be discussed.

## 1. Introduction

Cancer is the second leading cause of mortality in the European Union after cardiovascular diseases. In 2020, about 2.7 million people in the 27 EU countries were diagnosed with cancer, and nearly 1.3 million died from it [1]. Though large effort was made on cancer therapy, lives lost to cancer in the EU still tend to increase [2]. A major obstacle in cancer treatment is the development of resistance to many structurally dissimilar cytotoxic substances. This phenomenon is termed as multidrug resistance (MDR), which renders the cancer cell ineffective in accumulating drugs. It is related to activity of energy-dependent unidirectional, membrane-bound, drug-efflux transporter proteins, which belong to the ATP-binding cassette (ABC) transporter superfamily. In total, the human ABCome is composed of 48 genes, which encode seven subfamilies of ABC transporter proteins on the premise of sequence divergence and structural arrangement. The following nomenclature (alternative symbol in parenthesis) of the seven ABC transporter subfamilies was suggested by the Human Genome Organization’s Gene Nomenclature Committee (HGNC): ABCA(ABC1), ABCB(MDR), ABCC(MRP/CFTR), ABCD(ALD), ABCE(OABP), ABCF(GCN20) and ABCG(WHITE).

The first human ABC transporter, P-glycoprotein (P-gp, encoded by the MDR-1/ABCB1 gene), was characterized in 1976 [3]. It is also the most studied one regarding chemotherapy against cancer [4]. Subsequent to the discovery of P-gp, studies of cancer cells revealed other phenotypes, which showed multidrug resistance related characteristics. These multidrug resistance related proteins (MRPs) were later classified as the ABCC subfamily [5]. Simultaneously, a novel half transporter member of the ABC superfamily was identified from a resistant breast cancer cell line [6], hence named as breast cancer resistance protein (BCRP, encoded by the ABCG2 gene).

Additionally, there are multiple members of ABC transporter reported to export at least one anti-cancer agent, for instance ABCA2, ABCC2, ABCC3, ABCC4, ABCC5, ABCC6, ABCC11 [4]. Others, such as ABCB11 (also known as Spgp, sister of P-glycoprotein, or BSEP, bile salt exporter protein) which is predominantly expressed in liver, demonstrate also the potential to confer resistance to cytotoxic substrates like taxol and vinblastine [7,8]. Of all 48 human ABC transporter that have been described so far, the three aforementioned members ABCB1, ABCC1, ABCG2 are less organ specific [9] and were frequently observed with enhanced overexpression in multiple cancer types [10].

In principle, an ABC transporter comprises four domains, namely two transmembrane domains (TMDs) embedded in the lipid bilayer, and two nucleotide binding domains (NBDs) facing the cytoplasmatic space. The human ABC transporters can be either full- or half-transporters. In the full transporter transcript, the motifs are arranged as N-TMD-NBD-TMD-NBD-C, whereas the half ones have only one TMD and one NBD. Hence, the half-transporters should form homodimers or heterodimers to perform their function. The typical TMDs consist of 2 × 6 α-helices, which construct together one centralized efflux path for a variety of substrates. This folding pattern is conserved among the eukaryotic ABC transporters with, however, a lower sequence conservation comparing to the NBDs [11,12]. In the NBDs, where binding and hydrolysis of ATP take place, some motifs are highly conserved, such as Walker-A, Walker-B [13], the signature sequence (also as linker peptide or ‘LSGGQ’ motif), the Histidine loop [14], and the Glutamine loop [15].

With enhanced direct electron detectors [16] and image-processing packages [17], cryo-electron microscopy (cryo-EM) has elevated the progress of structural biology. This short review mainly focuses on those three ABC transporter members (ABCB1, ABCC1 and ABCG2), which are predominantly linked to the phenomenon of MDR in cancer therapy [18]. We summarize the research state from the structural biology aspect as well as structure-based in silico analyses. Finally, the impact of AlphaFold-2 database models on the understanding of the structural plasticity and flexibility of the aforementioned three ABC transporters is discussed.

## 2. ABCB1

P-glycoprotein (P-gp, P for permeability), by far the best-known and well-studied ABC-transporter, is composed of 1280 amino acids (170 kDa). In 2018, Kim et al. published for the first time a cryo-EM structure of human ABCB1 in an ATP-bound, outward-facing conformation, at a resolution of 3.4 Å (PDB code: 6C0V) [19]. Subsequently, Alam et al. obtained a series of three-dimensional (3D) structures of human P-gp via cryo-EM in a nucleotide-free state with resolutions from 3.58 to 4.14 Å. Remarkably, they managed to solve the structures both in substrate- (PDB code 6QEX, with taxol, a chemotherapeutic compound) and inhibitor-bound (PDB code 6QEE and 6FN1, with zosuquidar, a third generation ABCB1 inhibitor) states (Figure 1). These inward-open and occluded conformations demonstrate the plasticity of the central binding pocket [20]. More refined structures were reported by Nosol et al. in complex with the Fab fragment of an inhibitory monoclonal antibody MRK16 [21]. The highest resolution (3.2 Å) among their structures was retrieved with vincristine (an antitumor drug) in a substrate-bound occluded state (PDB code: 7A69). Moreover, they also determined additional structures (PDB code: 7A6C, 7A6E) bound with other potent inhibitors (elacridar and tariquidar) (Figure 1).

The broad substrate specificity of P-gp also refers to a crucial obstacle of designing specific ABCB1 inhibitors. Both Alam and Nosol observed multiple molecules in a connected region, comprising one substrate and two inhibitors. It was named by Nosol et al. as vestibule and access tunnel, which extends from the central substrate binding pocket to the cytoplasmic side [21] (Figure 1). The substrate binding pocket can embrace one molecule of substrate as well as one of the two inhibitors, while the second inhibitor stretches out and expands to the vestibule or even the access tunnel. Moreover, there were also reports about a potential transport of inhibitors (elacridar and tariquidar) under very low concentrations [22]. These observations suggest that there is no strict boundary between substrates and inhibitors, and the behavior might depend on the extent the small molecule occupies and interacts with the central cavity or beyond [21]. In general, it is worth noticing that the central binding site of human ABCB1 is mainly composed of aromatic side chains. In addition, the cytoplasmatic gate region (also being called access tunnel) participating in inhibitor binding is relative acidic. These observations are consistent with the fact that P-gp modulators are generally hydrophobic and positively charged or neutral.

## 3. ABCC1

When analyzing P-gp expression in different tumor types, it turned out that some cell lines show a high level of drug efflux but do not show upregulation of P-gp expression. In 1992, Cole et al. managed to clone a cDNA encoding a specific transporter protein of 170 kDa, and termed it multidrug resistance related protein 1 (MRP1) [5]. Later, it was grouped into the C-subfamily of ABC transporter through sequence analysis. According to HUGO Human Gene Nomenclature, the C-class human ABC protein gene family includes 12 members. Among them, nine are multidrug resistance related, namely MRP1-9 (or ABC1-6 and ABC10-12). Besides those, there is also the cystic fibrosis transmembrane conductance regulator (CFTR/ABCC7), which is an ATP-gated chloride channel instead of an active transporter [23]. The other two are sulfonylurea receptors SUR1 and SUR2 (or ABCC8, ABCC9, respectively), known as a component of an ATP sensitive potassium channel [24].

Topologically, the C-subfamily can be divided into “long” and “short” proteins. In general, the structural pattern of a typical ABC transporter remains, namely two TMDs and two NBDs. The only difference between the “long” and “short” ABCC proteins is an extra transmembrane domain 0 (TMD0, approximately 250 amino acids in addition) on the N-terminal of the long ones. This additional domain exists in 5 MRP members, namely MRP1, MRP2, MRP3, MRP6, MRP7 (ABCC1/2/3/6 and ABCC10, respectively), as well as in SUR1, SUR2. Interestingly, the TMD0 of SURs was firstly investigated by Chan et al. [25] and reported to mediate the protein–protein interaction, hence influencing but not being essential for the transport function of ABCC family members. Leukotriene C4 (LTC4, an endogenous substrate of MRP1 [26,27]) can still be transported without this domain [28], whereas certain mutations on it can reduce the transport activity [29,30]. Moreover, the loop 0 (L0) connecting TMD0 with TMD1, also called lasso motif in other MRPs [31], has been shown to engage in an intracellular recycling process [32] and transport activity [28], being probably even engaged in the gating process [33] of the two NBDs during the catalytic process.

At present, MRP1 is the first and best-studied ABCC/MRP protein in the field of clinical oncology [34]. Unfortunately, after almost 30 years since the discovery of ABCC1, the exact 3D structure of the human ABCC1 has not been successfully determined yet. Like P-gp, also MRP1 is capable of recognizing a diversity of structurally unrelated molecules. Moreover, as a full transporter, two subdomains of ABCC1 are also connected via a flexible linker. As alternatives, other eukaryotic homologous structures can be utilized to gain insights regarding the poly-specificity and the linker flexibility of human ABCC1.

A series of cryo-EM structures of ABCC1 from *Bos taurus* were published by the same research group from the year 2017 to 2020 (PDB code: 6UY0, 6BHU, 5UJ9, 5UJA) [35,36,37]. The bovine MRP1 (bMRP1) is close to the human transporter (hMRP1) both in sequence similarity (91%), and substrate affinity [34,38,39]. Based on this fact, analyzing the available bovine structures in different conformations (inward-facing, substrate-bound, outward-facing) can help to gain insights into hMRP1 structural determinants for substrate recognition and transport. Yet, none of the aforementioned experimental solved bMRP1 structures was captured with bound substrate or inhibitor, thus information is limited regarding the molecular features of the inhibitor/substrate-binding site. Nevertheless, known inhibitors and substrates of MRP1 are rather small and bipartite, which indicates that the binding pocket is composed of multiple positively charged and hydrophobic residues [38]. A ramification of this fact is that the whole transporting pathway of MRP1 is basic, which explains its trafficking preference of organic acids with large hydrophobic groups [40]. Regarding the linker between NBD1 and TMD2, some segments are present in cryo-EM structures (PDB code: 7MPE, 7M68, 7M69) of yeast cadmium factor 1 protein [41,42], (Ycf1p, a homolog of hMRP1) and cryo-EM structures of human CFTR (PDB code: 5UAK, 6MSM) [43,44]. Those structures revealed an interesting finding, that phosphorylation of this region leads to additional interaction between the linker and the peripheral face of the NBD1, hence allowing the two NBDs to dimerize.

## 4. ABCG2

Subsequent to the discovery of ABCC1, three different research groups independently found another multidrug efflux transporter. Originally it was named as breast cancer resistance protein (BCRP [6]), ABC transporter overexpressed in placenta (ABCP [45]), or mitoxantrone resistance (MXR [46]) protein, respectively. Based on sequence similarity and structural organization [47], it was assigned to the ABCG subfamily. The human G-class ABC protein subfamily is composed of six half transporters arranged as N-NBD-TMD-C, which is opposite to the other ABCs (TMD at the N-terminus). Among them, the second member ABCG2 is the one which confers resistance to several anticancer drugs [48,49].

ABCG2 is a membrane protein consisting of 655 amino acids (72 kDa). Its functional unit is minimal a homodimer, although reports about higher order homo-oligomers also have been published [50,51]. Nevertheless, high-resolution structures of human ABCG2 thus far are all determined as dimer [52,53,54,55,56,57]. As a half transporter with shorter sequence, ABCG2 differs to the other two aforementioned human multidrug transporters ABCB1 and ABCC1 with respect to domain-swapping and TMD-extension into the cytosol (Figure 2b,c). Consequently, it has a unique substrate capture and transport mechanism.

The first experimental solved full human ABCG2 occluded structure by Jackson et al. [53] in 2018 confirmed the intriguing finding of Taylor et al. [52] in 2017. They demonstrate an adjacent architecture of the NBDs under the inward facing occluded status, while the NBDs of the full transporters stay separated under this conformational state (Figure 2a). Orlando et al. [55] from the same group managed in 2020 to capture the resting ABCG2 in an antibody-, substrate- and inhibitor-free condition. The resting ABCG2 stabilizes in an inward facing apo conformation with the TMDs being separated as in the inward facing occluded conformation, but the two helices 5 shift towards the dimer axis to seal off the crevice in absence of bound inhibitor/substrate (Figure 2a). In comparison to that, both TMDs and NBDs of ABCB1 and ABCC1 adopt an open conformation to the cytoplasmatic side in the apo state (Figure 2b,c).

Overall, the group of Kaspar P. Locher has been publishing a variety of human ABCG2 structures through cryo-EM since 2017 (Table 1). Various conformational states (Figure 3) were captured including inward-facing (IF) apo (PDB: 5NJ3, 5NJG), inward-facing inhibitor/substate bound (PDB: 6FFC, 6ETI, 6FEQ, 6HIJ, 6HCO, 6VXH, 6VXI, 6VXJ, 7NEZ, 7NFD, 7NEQ), apo closed (PDB: 6VXF), and outward facing (OF) ATP bound states (PDB: 6HBU, 6HZM). Additional stable intermediate states were successfully solved, which were termed by Yu et al. [57] as turnover state 1 and 2. Both turnover states are inward-facing, with turnover-1 showing more widely separated NBDs and turnover-2 representing semi-closed NBDs and an almost fully occluded substrate cavity. As the transition from turnover 1 to turnover 2 seems to be the rate-limiting step, the relative reduced central binding cavity in turnover-2 was assigned to be the “test” size for whether the small molecule can be transported or not. Consistent with this idea, Kowal observed low-rate transport of a middle size molecule (tariquidar, originally developed as ABCB1 inhibitor), which can serve as a boundary example in the design of inhibitors for ABCG2.

Indeed, both substrates and inhibitors were determined in a predominantly overlapping cavity center, when superposing all occluded structures (Figure 4). Substrates, for instance estrone-3-sulphare (E1S), mitoxantrone (MXN), 7-ethyl-10-hydroxycamptothecin (SN38), topotecan and tariquidar, were found as one single molecule in the binding site (Figure 4c,d). Meanwhile, the same phenomenon of fitting two inhibitor molecules in the same cavity center as in multiple ABCB1 structures was also observed in ABCG2 structure 6ETI with two molecules of MZ29 (selective inhibitor of ABCG2 with neurotoxicity, a derivative of the fungal toxin fumitremorgin [58]). Moreover, comparing to the partial inhibitor MB136 (a derivative of tariquidar [59], a third-generation ABCB1 inhibitor), and imatinib [60,61,62] (Figure 4b, in cyan and rosa, respectively), which can fit only one molecule into that cavity, two molecules of MZ29 (Figure 4b, in light and dark blue) are able to reach out approaching the cytoplasmic membrane boundary to establish enhanced interactions. Yet, contact strength and molecular volume are not the only factor that differentiate an inhibitor from a substrate, as a mutation on Arg482 [63], which is outside the known binding site, can totally reverse the inhibition. Nevertheless, the known binding modes of substrates and (partial) inhibitors can still serve as example for the future design of ABCG2 modulators.

## 5. Protein Models and AlphaFold-2

Despite decades of investment from structural biologists, till now less than 10% of the human proteins have experimentally solved 3D coordinates deposited in the Protein Data Bank (PDB) (Protein-level coverage), which corresponds to approximately 17% of the sequences in the human proteome (residue-level coverage) [64]. However, this number could be considerably enhanced by creating protein homology models based on template structures from other species. When analyzing the structural landscape of ABC-transporter, we retrieved 160 available experimentally solved human structures from PDB covering 24 ABC-transporter members. For all ABC-transporters protein models can be found in the SWISS-MODEL repository. Under circumstances where no decent structure/model is available, all ABC transporters have suitable templates in the PDB for modeling with higher than 70% sequence similarity.

Briefly, this analysis is based on a KNIME [65] workflow which summaries the up-to-date status of the structural feasibility of all 48 ABC-transporter subfamilies. Upon a list of PDB identifier and entity ID, other experimental information, for instance method, resolution, sequence coverage, co-crystallized small molecules, and release data, are also simultaneously extracted by the workflow. If extra requirements appear, a threshold can be put on any of the aforementioned parameters. For instance, a 75% sequence coverage cut-off for the experimentally solved structures comparing to the respective full canonical sequence would filter out 59 structures and 3 transporter members, in which 91 are protein-ligand complexes (Figure 5a). Plotting the structure resolution value with the release time or ABC gen name reveals that (i) more than half of the structures (94 out of 160) were solved in the last three years (Figure 6), (ii) experimentally solved structures with lower than 3.5 Å resolution comprise 62% (99 out of 160) including both X-ray and cryo-EM solved structures (Figure 5b).

Finally, the availability of AlphaFold-2, an artificial intelligence approach for protein 3D structure prediction based on a neural network [66], had huge impact on the field. In the AlphaFold-2 database (https://www.alphafold.ebi.ac.uk/download (accessed on 10 December 2021)), 98.5% of all human proteins have a structural prediction. Therein, 58% of all modelled residues have a high (>70) confidence score (pLDDT, predicted local distance difference test score). This score scales between 0 to 100, in which a larger number indicates higher confidence on the prediction.

Taking the example of human ABCB1, despite the significant success in solving the overall structure, the communication between NBD and TMD remains unknown. The region between Ala631 and Pro693 connecting NBD1 with TMD2 is absent in all experimentally determined structures, but present in the AlphaFold prediction. It is known as a flexible linker being involved in substrate specificity and conformational change completing the full transport circle [67,68]. The group of dos Santos has been studying this region intensively via docking, homology modelling and molecular dynamics simulations [69,70]. Their latest update [71] on the linker secondary structure demonstrates a different configuration (available under this link: http://chemistrybits.com/downloads/systems/(accessed on 10 December 2021)) as it is shown in the AlphaFold-2 database, while both are in an inward-facing conformation. When superposed in Molecular Operation Environment (MOE) (Chemical Computing Group, 2019 software suite) [72], the AlphaFold prediction shows a higher expose level with the upper loop extending into the cytoplasmatic space, along with an extra helix segment on the lower part (Figure 7a), whereas the finalized model from dos Santos suggests a larger protein–protein contact surface with a more compact coil (Figure 7b).

Indeed, this linker domain is a highly flexible region. Along the years, models published by the dos Santos group have also been adjusted with the availability of murine and human P-gp crystal structures. Such intrinsically disordered regions (IDRs) are often depicted as very low confidence (pLDDT < 50) region in AlphaFold predictions. Regions with very low pLDDT score are suggested by Ruff et al. [73] no to be interpreted as structure but rather as a prediction of disorder. Hence, this part of the AlphaFold model cannot be taken as guidance for identifying the boundary of the inhibitor/substrate-binding site or potential interaction pattern variance between the linker and the TMDs.

Aside from the disordered region, defined secondary structures can be determined heterogeneous between experimentally solved structures and AlphaFold-2 database models. As an example, the TMD0 structure of ABCC1, which is found in bMPR1 structures published by Johnson et al. [35] in 2017 and Chen et al. [36] in 2019, as well as in Ycf1p structures solved by Bickers et al. [41] in 2021 and Khandelwal et al. [42] in 2022. The conformational transition across the transport cycle is shown to be mainly conducted by the TMDs and the NBDs, while the TMD0 undergoes a minor adjustment (Figure 8b). As TMD1 and TMD2 of AlphaFold-2 hMRP1, bMRP1 and Ycf1p structures superpose well with the respective inward facing conformations (bMRP1 PDB code: 5UJ9, Ycf1p PDB code: 7MPE), TMD0 from AlphaFold-2 hMRP1 and Ycf1p comprises another conformation leaning closer to the TMD1 (Figure 8a). This aligns with the finding of Bickers et al., who published the first Ycf1p structure in 2021. When they superposed the core of Ycf1p, bMrp1 and SUR1 structures, the TMD0s oriented magnificently different among the three structures [41].

The majority of the TMDs including TMD0 of AlphaFold-2 database models are predicted with high confidence value (pLDDT score > 70). In comparison, the bMRP1 structures consist of a less satisfactory local resolution (5 Å) of TMD0 [35]. The electro-density corresponding to this region is insufficient for detailed connectivity and residues assignment for every bMRP1 structure published thus far. Nevertheless, cryo-EM structures of Ycf1p demonstrate relative higher overall quality (3.2 Å–4.04 Å). Most importantly, these structures are solved with defined residue assignment of TMD0. However, 34% sequence similarity was found by applying the emboss needle global alignment on the TMD0 sequence of hMRP1 and Ycf1p. Such level of sequence similarity is sufficient for the comparison of overall structural packing. Yet, as a distant homolog, TMD0 of Ycf1p can interact with different regulatory partners. This leads to an ambiguous discrimination referring to the conformational variation of TMD0 in the AlphaFold-2 database model.

Notwithstanding the fact that TM0 is not directly engaged in the substrate transporting process, but influences the transport activity through long distance interaction, a conformational variance in this region can hardly be transferred into contributive information on the central cavity. Nevertheless, as supplementary to the existing limited polyalanine models, or models based on distant homologues, a high confidence model of the hMRP1 TM0 structure with conformational variance is not a drawback but an additional argument that the TMD0 can undergo a conformational shift while the rest of the structure stays relatively static.

The advantage of AlphaFold generating non-template-based full chain prediction, however, becomes a limitation in the case of polypeptide chains. As a half-transporter, only a monomer of ABCG2 is presented in the AlphaFold-2 database. As outlined above, a widely accepted concept for the ABCG2 functional unit is a homodimer [74,75], while other higher order oligomers are suggested to serve as regulators for the dimeric form [76]. Without a decent arrangement of the subunits, the value of the structural model is limited. On the one hand, the intermolecular disulfide bridges between the two halves could only be accessed in classical template-based homology models [77] before the crystal structure was solved in 2017. However, it was proven later that those intermolecular disulfide bridges are not essential for activity [78], and they might result from oxidation processes during sample preparation [76]. On the other hand, the central cavity for ligand as well as inhibitor binding can only be accessed in the dimeric form. It was observed in multiple cryo-EM structures, that substrates [54,56] (like estrone-3-sulphate, cholesterol, mitoxantrone, topotecan, tariquidar) and inhibitors [53] (Ko143, MZ29, and tariquidar-derivatives like MB136) bind in the same pocket on the symmetry axis. While inhibitors extend over the critical accommodate volume and establish additional interaction beyond, they seal the transporter in an inward facing conformation. With one monomer structure from AlphaFold prediction, neither the intermolecular interaction nor the binding pocket composition can be illuminated. Moreover, conformational transformation during the transport circle requires also the full functional formation. Consequently, even when the DeepMind team launched a newly developed AlphaFold-multimer protocol [79], it still requires critical assessment to utilize the prediction appropriately.

Nevertheless, DeepMind released also the whole algorithm framework of AlphaFold-2 in GitHub [66], with which one can further explore other possibilities, such as generating structural predictions with chosen templates, as well as yielding new folding patterns. Above all, the novel deep learning framework developed by DeepMind has still the potential to reverse-feed structural biology by providing predicted model fitting into cryo-EM density [80] or by molecular replacement to solve X-ray structures [81]. Hence, it might accelerate the determination process of, e.g., the human ABCC1 structure. Even when an experimental determined structure is available, like ABCB1 and ABCG2, computational prediction can still provide valuable information, such as signaling the disorder level of a region, or presenting the structure in another conformation. Both accelerating experimental structure-solving and providing structural models in other conformations or packing format are valuable contributions that AlphaFold can provide under respective critical perception.

## 6. Conclusions

Reversing MDR remains a major challenge in cancer chemotherapy. The ABC transporter members discussed in this short review are the most studied targets towards unraveling the mechanism of drug resistance in the chemotherapy of multiple cancer types. In the past decades, the research on these proteins was enormously hampered by the fact that membrane proteins are extremely troublesome to extract, purify, and crystallize. However, considerable progress was made by the structural biologists. This comprises not only solving 3D coordinate experimentally, but also generating reliable predictions. Two out of these three aforementioned human ABC transporters have multiple cryo-EM structures along the transition time line. Only human ABCC1 has no experimentally solved structure so far, but available homologue structures can still serve as valuable templates to model the human protein. This review summarizes the current state of structural study both experimentally and theoretically on these three transporters. With special focus on the binding range of substrate versus inhibitor, the gained knowledge can shed light on the understanding of the boundary of substrates versus inhibitors. In the case of ABCC1, the main obstacle is the absence of an experimentally solved structure. Nonetheless, homolog structures and AlphaFold model presented in this review are a precious resource, which can not only deliver relevant information regarding substrate specificity, but also potentially accelerate the solving of human protein structures.

## Figures and Tables

**Figure 1 molecules-28-00495-f001:**
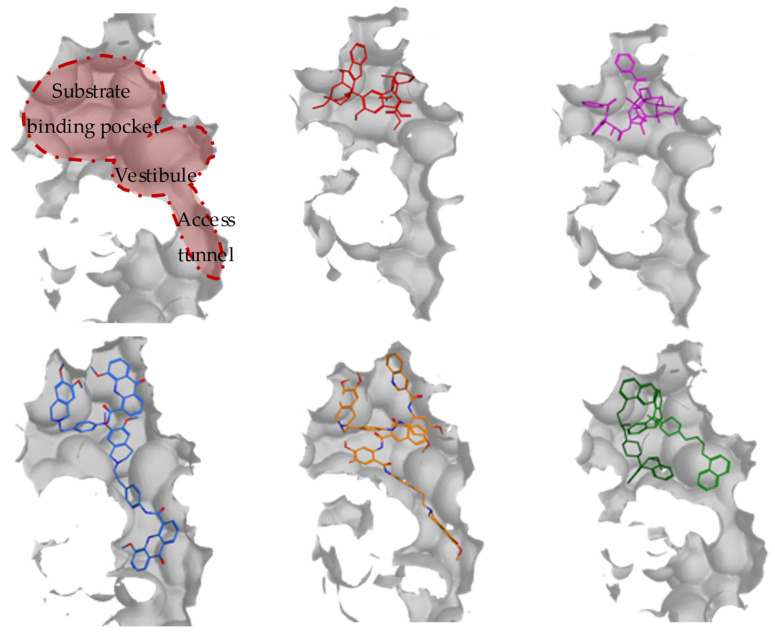
The Interaction surface of various ABCB1 cryo-EM structures with their bound substrates or inhibitors. The substrate binding pocket, vestibule and access tunnel are marked out in a red area, while one molecule of substrate vincristine (in red, from 7A69), taxol (in rosa, from 6QEX) and two molecules of inhibitors elacridar (in blue, from 7A6C), tariquidar (in orange, from 7A6E), zosuquidar (in green, from 6QEE) are shown as sticks.

**Figure 2 molecules-28-00495-f002:**
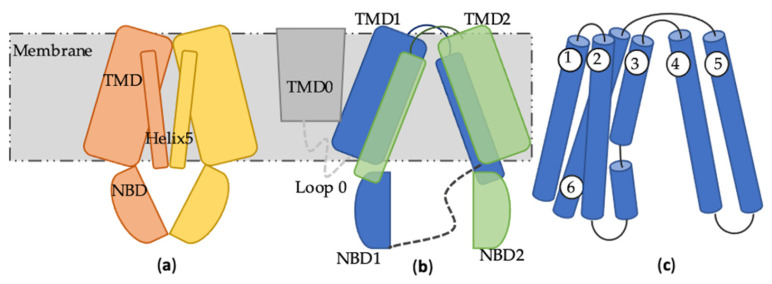
Inward facing apo conformation comparison of ABCG2 dimer with ABCB1 and ABCC1. (**a**) the two helices 5 from two ABCG2 subunits (one monomer in orange, another in yellow) rotate towards the central crevice, while the remaining helices of the NBD rotate towards the intracellular space. (**b**) ABCB1 and ABCC1 are full transporter with two TMDs and NBDs (TMD1 and NBD1 in blue, TMD2 and NBD2 in green), while the ABC1 comprises an extra TMD0 colored in grey connected by loop 0. (**c**) Topology diagram of TMD1 domain-swapping of full transporters with two helices (helix 4 and helix 5) cross-over the central axis.

**Figure 3 molecules-28-00495-f003:**
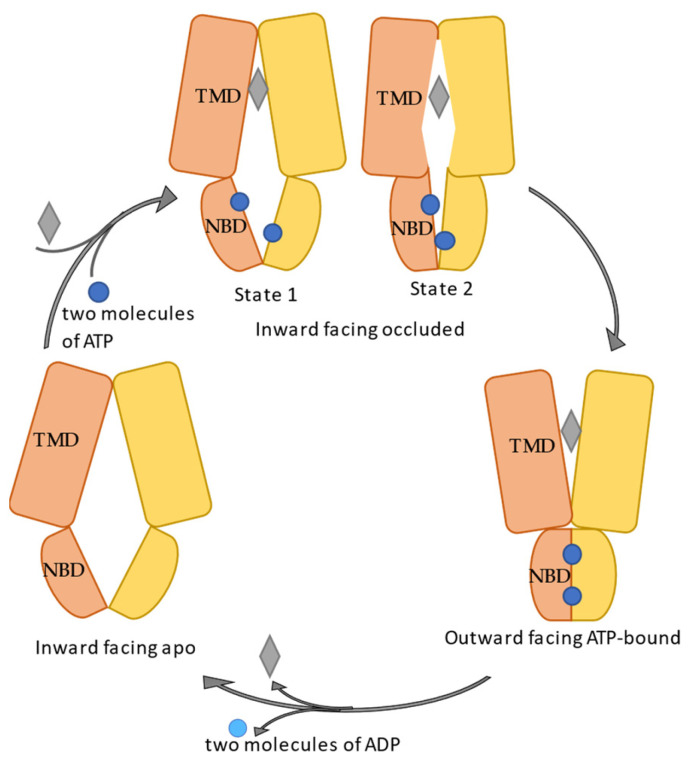
Schematic illustration of the conformational change during the transport cycle of ABCG2. Orange and yellow cartoons represent two monomers, each with one TMD and one NBD. Dark blue spheres are ATP molecules, light blue spheres represent ADP, while the substrate is shown as grey rhombus.

**Figure 4 molecules-28-00495-f004:**
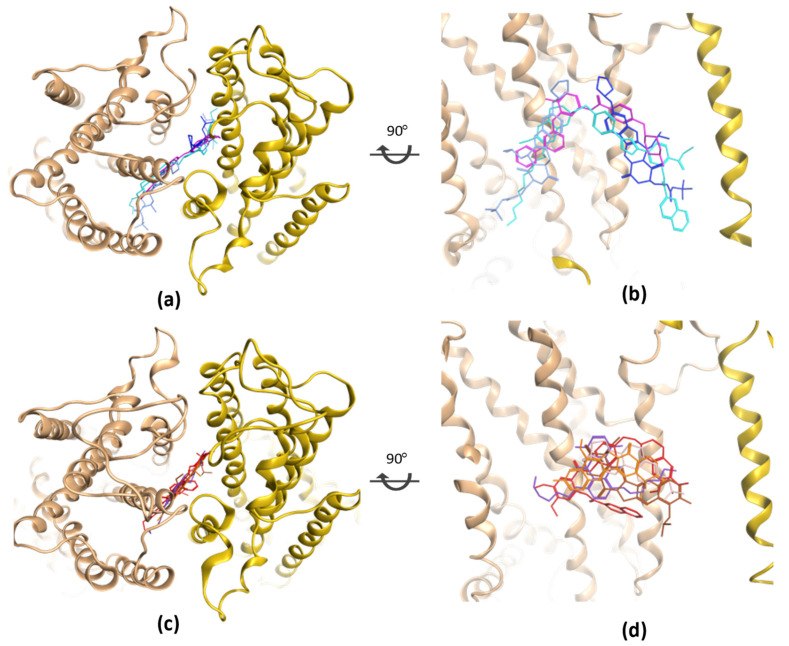
Superposition of bound inhibitors and substrates in the cavity center of ABCG2 cryo-EM structures. (**a**) Inhibitors MZ29×2 (in light and dark blue), MB 136 (in cyan), and imatinib (in rosa) from different PDB structures 6ETI, 6FEQ, 6VXH overlay in the same cavity center. (**c**) Overlay of 5 substrates in the cavity center of multiple ABCG2 cryo-EM structures, namely SN38 (in orange, from 6VXJ), E1S (in pink, from 6HCO), tariquidar (in red, from 7NEQ), topotecan (in brown, from 7NEZ), and mitoxantrone (in violet, from 6VXI). (**b**) and (**d**) are the cross-section view of (**a**) and (**c**), respectively.

**Figure 5 molecules-28-00495-f005:**
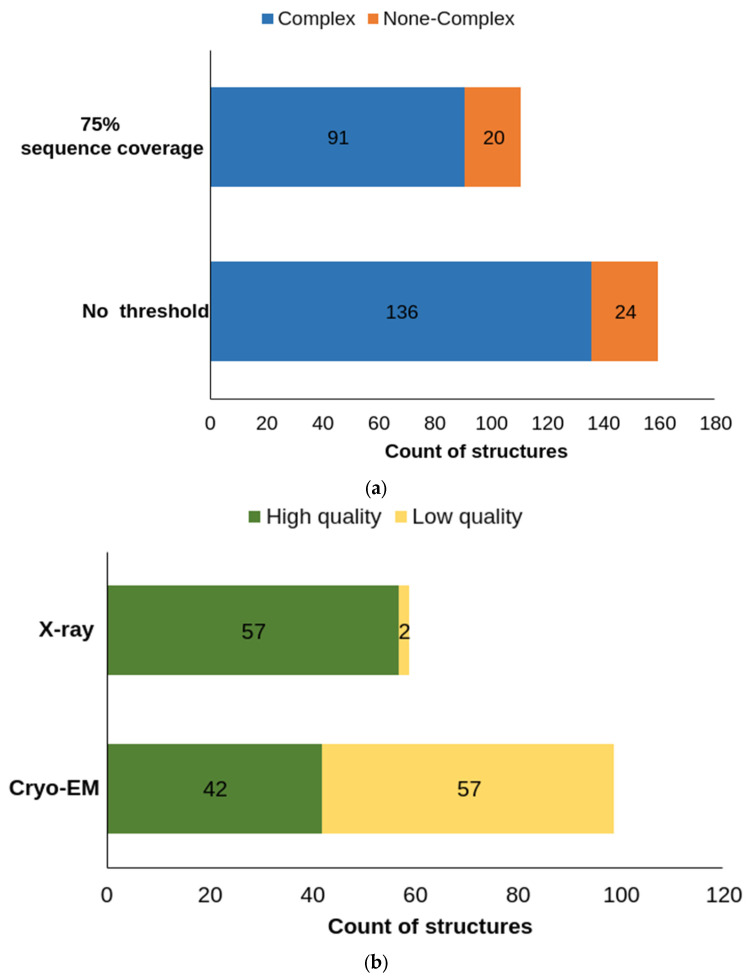
Composition of PDB structure. (**a**) in total, 160 PDB structures are retrieved for all 48 ABC-transporters, in which 136 concatenates (blue fraction) contain crystalized small molecules, 24 not (orange fraction). A threshold of 75% sequence coverage filters out 59 structures, leaving 101 experimentally solved human ABC structures. Among these, 20 are without and 91 are with a small molecule complexed. (**b**) For the total 160 PDB structures, 99 structure possess a resolution value lower than 3.5 Å (green fraction), among which 57 are solved by X-ray, and 42 are from cryo-EM. Apart from these two methods, one structure was solved by NMR, which is not included in the bar chart.

**Figure 6 molecules-28-00495-f006:**
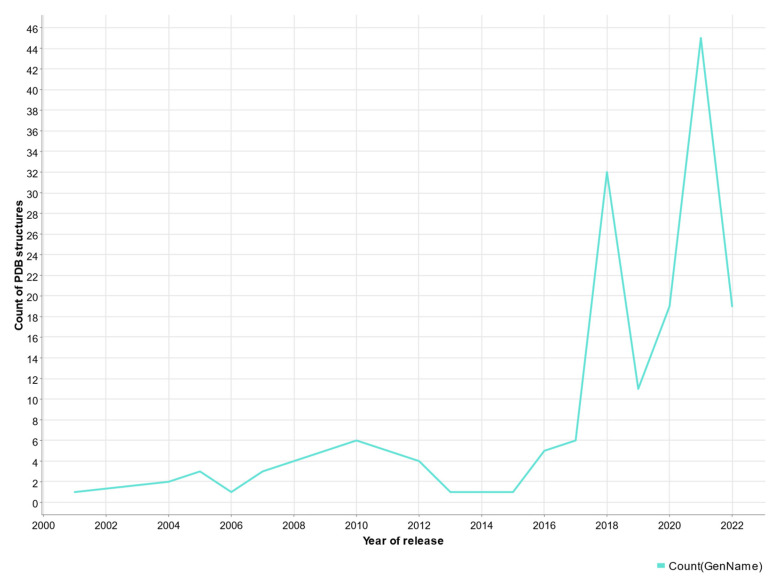
Plot of the number of experimentally solved structures vs. the year of release.

**Figure 7 molecules-28-00495-f007:**
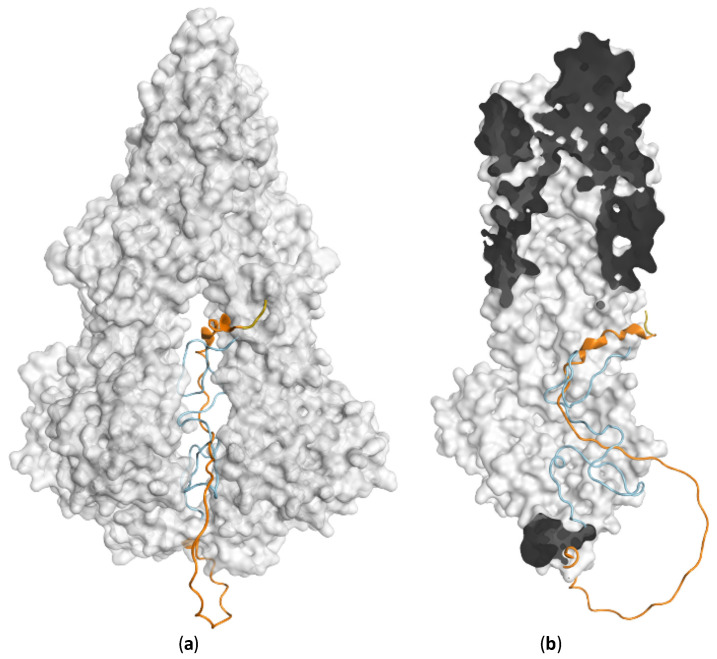
Flexible linker configuration between NBD1 and TMD2 of P-gp. Linker of AlphaFold-2 prediction colored in orange, indicating a low confidence pLDDT value. In comparison to the AlphaFold position, the model from dos Santos group packs the linker (in blue) tighter in the crevice. (**a**) Linker position in the interface of P-gp (linker from AlphaFold model in orange, linker from dos Santos group in blue). (**b**) The folding of the the linker viewed from the interface.

**Figure 8 molecules-28-00495-f008:**
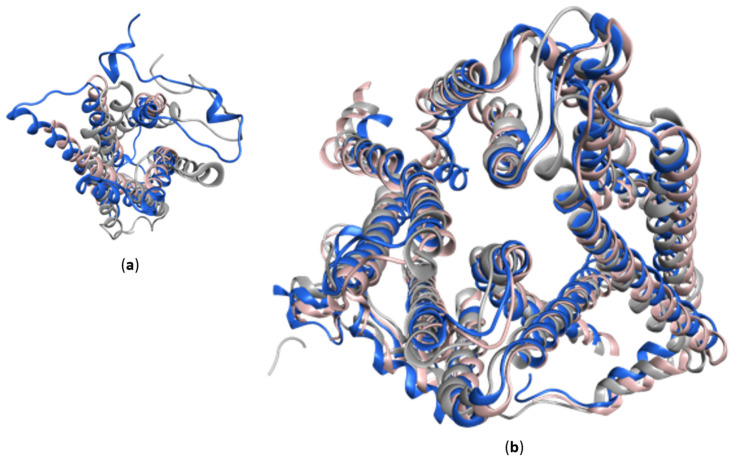
Conformational arrangement of bMRP1 (PDB code: 5UJ9), hMRP1 AlphaFold-2 model and Ycf1p (PDB code: 7MPE) viewing from extracellular side in MOE. (**a**) Under superposition of the core TMDs, TMD0 of AlphaFold-2 model and Ycf1p undergoing another conformational derivation. (**b**) Superposition of the TMD1 and TMD2 of three inward facing structures (bMRP1 in pink, AlphaFold-2 hMRP1 model in blue, Ycf1p in grey).

**Table 1 molecules-28-00495-t001:** Summary of cryo-EM structures of human ABCG2. All structures collected in this stable are exclusively solved via cryo-EM.

PDB	Conformation	Cocrystal Molecules	Property	Resolution (Å)	References
5NJ3	IF (NBDs modeled)	-	-	3.78	Taylor et al., 2017 [52]
5NJG	IF (Only TMDs)	-	-	3.78	Taylor et al., 2017 [52]
6FFC	IF	MZ29 × 2	Inhibitor	3.56	Jackson et al., 2018 [53]
6ETI	IF	MZ29 × 2 (Fab)	Inhibitor	3.10	Jackson et al., 2018 [53]
6FEQ	IF	MB136 (Fab)	Inhibitor	3.6	Jackson et al., 2018 [53]
6HIJ	IF	MZ29 × 2 (Cholesterol and phospholipid surrounded)	Inhibitor	3.56	Jackson et al., 2018 [53]
6HCO	IF	E1S	Substrate	3.58	Manolaridis et al., 2018 [54]
6HZM	OF	ATP × 2 + Mg^2+^ (Alternative placement)	-	3.09	Manolaridis et al., 2018 [54]
6HBU	OF	ATP × 2 + Mg^2+^	-	3.09	Manolaridis et al., 2018 [54]
6VXF	Apo-closed	-	-	3.50	Orlando et al., 2020 [55]
6VXH	IF	Imatinib	Inhibitor	4.00	Orlando et al., 2020 [55]
6VXI	IF	Mitoxantrone	Substrate	3.70	Orlando et al., 2020 [55]
6VXJ	IF	SN38	Substrate	4.10	Orlando et al., 2020 [55]
7NEZ	IF	Topotecan (Fab)	Substrate	3.39	Kowal et al., 2021 [56]
7NFD	IF	Mitoxantrone (Fab)	Substrate	3.51	Kowal et al., 2021 [56]
7NEQ	IF	Tariquidar (Fab)	Substrate	3.12	Kowal et al., 2021 [56]

## Data Availability

Data is contained within the article.

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
