# Peer review of "A Structure-Based View on ABC-Transporter Linked to Multidrug Resistance"

_molecules, 2023, doi:10.3390/molecules28020495_

Round 1

Reviewer 1 Report

In this brief review, the authors have discussed the recent structural work on the multidrug resistance-linked ABC transporters. In addition, they have provided some information about the artificial intelligence (AI)- based AlphaFold-2 and its impact on the understanding of the structural aspects of some of the ABC transporters including ABCB1, ABCC1 and ABCG2. Although this brief review is well presented, one major concern is that it lacks important information about the substrate-binding pocket of these transporters, and the broad substrate specificity.

Specific comments:

1.      Several structures of ABCG2 (Table 1) and ABCB1 bound to substrates and inhibitors have been published. In addition, the LTC4-bound structure of bovine ABCC1 is available. Structural information about the the drug-binding pockets of ABCG2, ABCB1 and ABCC1 should be included in this review. This is very relevant to the discussion of their structural aspects, as two molecules of inhibitors whereas only a single molecule of substrates binds in the pocket of ABCB1. On the other hand, even though there is a significant overlap between the substrate specificity of ABCB1 and ABCG2, only a single molecule of either a substrate or an inhibitor is found in the binding site of ABCG2. In the case of ABCC1, a single molecule of LTC4-GSH is bound to the pocket. The authors also need to discuss whether the AI-based AlphaFold-2 could shed some light on the polyspecificity of these transporters.

2.      The structure of ABCB11 (BSEP) is also available. This transporter under certain conditions has been shown to confer resistance to anticancer drugs. This is also not discussed by the authors.

3.      Minor point-p2, line 53— “linker peptide or Walker-C motif”. It is incorrect to call the dodecapeptide signature region “Walker C”, because G. Walker only described the Walker A and Walker B motifs in Fo-f1 and other ATPases. The signature sequence is only found in ABC transporters. Some of investigators refer this region as the “C region”.

Author Response

Thank you for the kind review and constructive suggestions. Please find more detailed response in the uploaded file.

Reviewer 2 Report

This paper presents a review on 3 types of ABC transporters involved in multi-drug resistance: ABCB1 (or P-glycoprotein), ABCC1 (or MRP1), and ABCG2. The authors introduce these three proteins and also their importance. However, the discussion is incomplete and needs additional information to aid the expert and non-expert in understanding how structures of these ABC transporters has yielded insight into multidrug resistance and what the unanswered questions are, which I believe was their intent.

First, the reader would benefit from a schematic diagram of the ABC transporters they discuss: (1) ABCB1 (or P-gp), including the disordered region linking TMD1 and NBD1; (2) ABCC1 (or MRP1); and ABCG2. The diagram can be used to highlight the domain-swapping in ABCB1 and ABCC1 that is absence in ABCG2 that the authors discuss later. This diagram would be very useful for non experts. Additionally, the authors are not consistent in the their abbreviations - NBD vs NB, TMD vs TM when describing the sequence of domains in ABC proteins.

Second, the discussion of the structures of the different ABC proteins bound to different ligands would benefit from some figures highlighting the interactions they refer to in the text.

The authors need a more complete explanation of what is known about TMD0 in the ABCC family of proteins, and what is also known about the NBD1-TMD2 linker in ABCC proteins. First, the TMD0 structure was first elucidated with the structures of KATP channels - the ABCC proteins SUR1 and SUR2 form the regulatory subunits of these channels. Secondly, while, as the authors state, TMD0 of MRP1 could only be modeled as a poly-Ala, the atomic model could be elucidated from the 2021 structure of Ycf1p, the yeast homologue of the protein, and confirmed by the later (2022) Ycf1p structure. Finally, it is well-known that ABC proteins, like other membrane proteins, contained disordered regions. Further, much work has been to characterize the NBD1-TMD2 linker of the ABCC protein CFTR. This should be mentioned in the paper.

In the discussion of ABCG2, the authors allude to different conformations: inward-facing occluded states vs inward-facing state. A figure highlighting these and other states of the protein would be beneficial. A number of structure papers that capture bacterial ABC transporters in different states can guide the writing of this discussion. Also, the authors need to re-word the description of the inward-occluded state on lines 151-153, as it hard to understand, especially without a figure.

The authors also present an analysis regarding ABC transporter structures in the PDB relating various parameters: e.g., and the time since release and resolution, or whether small molecules (assume substrates or nucleotides) are bound. While it gives a good idea into how many structures are available, the authors should be careful when comparing different techniques - i.e. Xray and cryo-EM given the fact that at the same resolution, more information is available from cryo-EM maps that Xray maps.

Finally, the paper requires serious edits for grammar and punctuation. For example, the sentence on line 195 and 196 is not a complete sentence. The sentence starting on line 200 needs to start with a capitalized For.The sentence spanning lines 24-26 is difficult to understand and may be better as two sentences. I believe that the sentence starting on line 85 is missing the word "not", as in " ... but not to which ...". These are just some examples.

Author Response

We sincerely appreciated the feedback and carefully worked on the manuscript. Please find in the attached file more details regarding the points mentioned in the review report. 

Round 2

Reviewer 1 Report

The revised manuscript is significantly improved. The authors have addressed my concerns adequately by incorporating new information in the revised manuscript.